# New Pyridone-Based Derivatives as Cannabinoid Receptor Type 2 Agonists

**DOI:** 10.3390/ijms222011212

**Published:** 2021-10-18

**Authors:** Manuel Faúndez-Parraguez, Carlos Alarcón-Miranda, Young Hwa Cho, Hernán Pessoa-Mahana, Carlos Gallardo-Garrido, Hery Chung, Mario Faúndez, David Pessoa-Mahana

**Affiliations:** 1Pharmacy Department, Faculty of Chemistry, Pontificia Universidad Católica de Chile, Vicuña Mackenna 4860, Santiago 7820436, Chile; ctalarcon@uc.cl (C.A.-M.); aycho@uc.cl (Y.H.C.); carlos.gallardo@ug.uchile.cl (C.G.-G.); hchung@uc.cl (H.C.); mfaundeza@uc.cl (M.F.); 2Organic Chemistry and Physical Chemistry Department, Faculty of Chemical and Pharmaceutical Sciences, Universidad de Chile, Olivos 1007, Santiago 7820436, Chile; hpessoa@ciq.uchile.cl

**Keywords:** cannabinoids, 2-pyridone, synthesis, CB2R agonists

## Abstract

The activation of the human cannabinoid receptor type II (CB2R) is known to mediate analgesic and anti-inflammatory processes without the central adverse effects related to cannabinoid receptor type I (CB1R). In this work we describe the synthesis and evaluation of a novel series of N-aryl-2-pyridone-3-carboxamide derivatives tested as human cannabinoid receptor type II (CB2R) agonists. Different cycloalkanes linked to the N-aryl pyridone by an amide group displayed CB2R agonist activity as determined by intracellular [cAMP] levels. The most promising compound **8d** exhibited a non-toxic profile and similar potency (EC50 = 112 nM) to endogenous agonists Anandamide (AEA) and 2-Arachidonoylglycerol (2-AG) providing new information for the development of small molecules activating CB2R. Molecular docking studies showed a binding pose consistent with two structurally different agonists WIN-55212-2 and AM12033 and suggested structural requirements on the pyridone substituents that can satisfy the orthosteric pocket and induce an agonist response. Our results provide additional evidence to support the 2-pyridone ring as a suitable scaffold for the design of CB2R agonists and represent a starting point for further optimization and development of novel compounds for the treatment of pain and inflammation.

## 1. Introduction

The endocannabinoid system comprises a complex network of lipid signaling mediators in which different proteins participate in the modulation of numerous physiological and pathophysiological processes [1,2]. Cannabinoid receptor type II (CB2R) belongs to the family of heptameric receptors coupled to G proteins (GPCRs). This receptor was identified and cloned from HL60 cells [3] and was initially considered as ‘peripheral cannabinoid receptor’ due to its wide distribution in peripheral cells and tissues, particularly in those of the immune system [4,5]. However, later studies showed its expression also within the Central Nervous System (CNS) especially under states of inflammation [4,6]. Various studies have shown that the activation of CB2R can block activation of microglia cells but has little effect on the normal functioning of neurons within the CNS [7,8]. Several reports indicate that the activation of CB2R is analgesic and CB2R agonists have been shown to suppress responses in animal models of both acute and neuropathic pain [5,9,10]. Additionally, cannabinoids have well-established anti-inflammatory properties and recently, effects in the gut-lung-skin barrier epithelia have been reported showing promising results in in vitro and in vivo animal studies [11]. Furthermore, the endocannabinoid system is intimately related to neurological function and neurodegenerative diseases with animal models studies showing beneficial effects for the treatment of brain injuries and multiple sclerosis [12]. Therefore, CB2R agonists represent potential alternatives for the treatment of pain and inflammation both in the peripheral and CNS [13].

The discovery of the CB2R directed research efforts towards the understanding of its role and action. Several reports on the structural requirements for ligand binding to the receptor led to the discovery of many different families of cannabinoid ligands including classical cannabinoids structurally related to THC, eicosanoids analogous to endocannabinoids and synthetic cannabinoids, most of the latter being heterocycles, aminoalkylindoles (represented by WIN55212-2), arylpyrazoles, quinolones and pyridone carboxamide derivatives.

Heterocyclic compounds represent an important source of pharmacologically active molecules and more than 85% of all biologically active compounds contain heterocyclic scaffolds [14]. They are frequently used to alter physicochemical properties of molecules such as lipophilicity, polarity and hydrogen bonding capacity which can improve the pharmacodynamic and pharmacokinetic profile [15]. The pyridone heterocycle is a 6-membered aromatic ring with a carbonyl group and a nitrogen heteroatom which has found great use in drug discovery strategies [16]. Relevant characteristics associated to this structure have been described by Y. Zhan and A. Pike, such as its ability to act as both a hydrogen bond acceptor and donor; act as a bioisostere of amides, phenyls and other nitrogen and oxygen-containing heterocycles, and the capacity to modulate the lipophilicity, solubility, and metabolic stability [16].

Previous reports have explored the 2-pyridone scaffold in the cannabinoid system particularly in the CB2R with promising results (Figure 1) [17,18,19,20,21,22]. Kusakabe et al., reported a 2-pyridone-based compound displaying high CB2R affinity and selectivity. They proposed that the pyridone scaffold could provide optimal lipophilicity for the design of CB2R ligands and predicted possible hydrophobic interactions with W194 and F117 [19].

Additionally, the recently reported Cryo-EM structure of human CB2R bound to the selective agonist [23] revealed important insight into the lipophilic binding cavity and provided structural determinants to distinguish CB2R agonists from antagonists. Antagonist extension deeper into the binding cavity that enables interaction and rearrangement of the conserved residue W258^(6.48)^ was proposed to be a critical feature of antagonist binding. According to their findings, the three residues, W194, F117 and W258 play an important role in distinguishing agonist and antagonist response together with ligand efficacy [23]. Therefore, rational design of CB2R agonists considering interactions in the three described cavities of the orthosteric binding site but avoiding contacts with W258 could be used to develop new CB2R agonists.

In an effort to identify novel CB2R agonists, in the present work we report the synthesis, evaluation and molecular docking study of two series of pyridone derivatives with the aim of initiating a SAR exploration around the pyridone central scaffold and identify new high affinity pyridone-based derivatives as CB2R ligands. Based on previously reported ligands and the described “three-arm pose” of CB2R binding ligands, different cycloalkyl and cycloaryl substituents were explored around the pryridone ring. Functional activity was evaluated through determination of intracellular cAMP and molecular docking studies were carried out to rationalize binding site interactions.

## 2. Results and Discussion

### 2.1. Chemistry

All compounds were synthesized as shown in Figure 1. Firstly, three N-aryl-4,6-dimethyl-2-oxo-1,2-dihydropyridine-3-carboxylic acids were synthesized from compound **3** using different substituted anilines (step c, Figure 1) to obtain the corresponding substituted amides **4a**, **4b** and **4c**. These amides were cyclized using acetylacetone and piperidine as a catalyst to yield compounds **5a**, **5b** and **5c**. The pyridone derivatives were then hydrolyzed using potassium hydroxide in ethanol 80% under reflux heating obtaining the carboxylic acids derivatives **6a**, **6b** and **6c**.

It is noteworthy to mention that hydrolysis of the nitrile derivatives proved to be harder than expected and both acid and basic conditions were studied. As described in Figure 2, different side products were obtained depending on the reaction conditions. Only decarboxylated product **9a** was obtained under acidic medium while under basic medium, the obtained product depended upon the reaction temperature. Heating the reaction below 100 °C stopped the reaction at the amide intermediate **10a** whereas heating the reaction above 100 °C completely hydrolyzed the precursors to product **6a** (95% relative yield). 

The carboxylic acid derivatives **6a**, **6b** and **6c** were finally reacted with different cycloalkyl amines (cyclohexylamine, cycloheptylamine and 1-adamantylamine), according to Figure 3. The products **7** and **8** were obtained using the same synthetic procedure whereby the respective amines were coupled to the carboxylic acid in the presence of BOP as coupling reagent. The relative yield for compounds **7** varied between 50% and 70% and for compounds **8** varied between 70 and 80%.

### 2.2. Human CB2R cAMP Assay (Agonism Effect)

Functional activity of the synthesized compounds was evaluated through their ability to decrease the accumulation of intracellular cAMP levels (Eurofins Cerep services), [24] and the results are displayed in Table 1. The results show a dependence of activity on the nature of the group present in position X. Heteroaryl derivatives presented little to no activity in contrast to cycloalkyl derivatives with three of the compounds (**8b**, **8c** and **8d**) showing activity above 30%. No significant activity was observed in the 2-benzothiazole derivative compounds (**7a** and **7g**) while **8d** showed the highest agonist response and the EC50 was determined to be 112 nM (Figure 2).

Regarding the cycloalkyl-substituted derivatives, the presence of an adamantyl group seemed to favor activity over the cycloheptyl ring (**8c** and **8d** vs. **8b**). Compound (**8a**) showed the lowest percentage of agonist response and followed by the cycloheptyl derivative compound (**8b**). Replacement with a bulkier adamantyl group, the percentage of human CB2R agonist response increased over 50% at concentration 10 µM (**8c**).

Comparing compounds with an adamantyl group in position X, activity was also influenced by the substituent in position Y. A *p*-tolyl group presented maximum response (**8d**) whereas a phenyl group (**8c**) showed half the maximal response at the same concentration (10 μM), suggesting that polar groups in this region are not favorable for activity.

### 2.3. Molecular Docking Studies

Molecular docking studies on the CB2R predicted that the designed ligands bind in the transmembrane (TM) region defined by TM2-TM3 and TM5-TM7 in a similar disposition as that shown by WIN55212-2 in the Cryo-EM structure of human CB2R [23]. 

Based on reported 3D structure data, three distinct cavities in the binding pocket can be identified which accommodate the three-group scaffold of the agonist WIN55212-2 and antagonist AM10257, in what has been described as a “three-arm pose” [23,25]. According to the interactions established by WIN55212-2 and AM10257, respectively, we define as cavity 1 by residues F87, F91 and F94 which form a hydrophobic pocket that binds the naphthyl or adamantly moiety, referred to as arm 1; cavity 2 by residues I186, W194 and M265 which participate in hydrophobic interactions with the morpholine moiety or hydroxypentyl chain, referred to as arm 2; and cavity 3 defined by residues F117, V113, F183, F281, W258 and V261 that establish π-π stacking and hydrophobic interactions with the oxazinoindole or pyrazole core. Here a downward extension of a longer group towards W258 in the antagonist structure defines arm 3 (Figure 3A). 

Our results showed that the most active compound **8d** stabilized in the orthosteric site forming an “L-shape” pose (Figure 3B). The pyridone core acted as a central scaffold within the center of the binding site and cavity 3 engaging in hydrophobic interactions with V113, V261 and M265 and π-π interactions with F117 and F183. As we expected, the 2-pyridone moiety acted as a pivot directing the substituent groups to the binding cavities. The adamantyl group (arm 1) extended towards cavity 1 interacting with F94, H95 and I110, while the p-tolyl group (arm 2) was oriented towards cavity 2 were aromatic and hydrophobic interactions with residues Y190, L191 and W194 were possible (Figure 3B).

Furthermore, Hua et al., have also reported the crystal structure of CB2R bound to the selective agonist AM12033 with a resolution of 3.2 Å (PDB:6KPC) [26]. Even though AM12033 is structurally different to WIN-55212-2 and **8d**, all of them share a common “L-shape” conformation in the orthosteric site [23,26]. In all cases, a hydrophobic central core occupies part of cavity 3 and connects an arm 2 side chain in cavity 2 with a voluminous and hydrophobic arm 1 that extends towards cavity 1 (Figure 4B). 

Within the same general scaffold, the steric effect of an additional substituent group in cavity 3 was found to be critical in distinguishing agonists from antagonists. Inverse agonism activity was observed when deep insertion of longer sidechains in cavity 3 allowed strong π-π interaction (3.0 Å) with W258 which plays a key role in receptor activation/inactivation (Figure 5b) [23,26]. Superposition of the predicted binding pose of **8d** with agonist WIN 55,212-2 and antagonist AM10257 showed that **8d** does not reach residue W258 (>5 Å) consistent with the observed agonist profile in functional assays (Figure 5 and Figure 6).

### 2.4. Cell Ciability Assays

To evaluate the effect of compound **8d** on cell viability and characterize its toxicity profile, neutral red uptake assay was performed as described in Methods section. Two different cell lines were used: HepG2 as a model of hepatic tissue; and HL-60 cells as a model of blood cells. Low levels of expression of CBR exists in normal liver [27] however, several pathological conditions have shown high upregulation of hepatic CB2R [28].

As shown in Table 2, **8d** did not show significant effect on cell viability at concentrations as high as 200 μM in HepG2 cells when compared to control (IC_50/HepG2_ >200 μM). In the case of HL-60 cells, exposure to **8d** at 100 mM retained viability to approximately 80% although at 200 μM viability was reduced to 10% (IC_50/HL-60_ >100 μM). Considering the EC_50_ value of **8d** on CB2R (0.11 μM), there is at least 890 times difference with respect to the toxic concentration of the same compound highlighting the low toxicity profile of **8d**.

## 3. Materials and Methods

### 3.1. Chemistry

All organic solvents used for the synthesis were analytical grade. Melting points were determined on a Stuart Scientific SMP3 apparatus. ^1^H and ^13^C NMR spectra were obtained on a Bruker AM-400 spectrometer. The chemical shifts are expressed in ppm (δ scale) downfield from TMS, J value are given in Hertz for solutions in CDCl3 unless otherwise indicated. Column chromatography was performed on Merck silica gel 60 (70–230 mesh). Thin layer chromatographic separations were performed on Merk silica 60 (70–230 mesh) chromatofoils.

#### 3.1.1. General Procedure for the Synthesis of Cyanoacetic acid Hydrazide (**2**)

Cyanoacetic acid hydrazide was obtained by careful addition of 3.2 mL (30 mmol) of ethyl cyanoacetate to a solution of 1.8 mL (60 mmol) of hydrazine hydrate in ethanol (20 mL) with stirring at 0 °C. Then the mixture was heated to room temperature with stirring at room temperature for 2 h. The solid formed was filtered, washed with diethyl ether and dried obtain 2.82 g of product. Yield = 95%. White solid. This compound was used without previous purification at the next synthetic step.

#### 3.1.2. General Procedure for the Synthesis of 1-Cyanoacetyl-3,5-dimethylpyrazole (**3**)

2.8 g (28 mmol) of cyanoacetic acid hydrazide was dissolved in 100 mL of distilled water. To the mixture was added 1 mL of concentrated HCl. The solid was completely dissolved and then acetylacetone 2.93 mL (28 mmol) was added drop by drop with vigorous stirring at room temperature for 5 h. The resulting mixture was filtered and the solid obtain was washed with cold diethyl ether obtaining 3.38 g of white solid. Yield = 74%. m.p. 119–121 °C (CDCl_3_, 400 MHz) δ 2.24 (s, 3H, CH_3_), 2.54 (s, 3H, CH_3_), 4.30 (s, 2H, CH_2_-CN), 6.04 (s, 1H, pyrazole-H).

#### 3.1.3. General Procedure for the Synthesis of N-Aryl-2-cyanoacetamide (**4a–c**)

Equimolar amounts (6.9 mmol) of 1-cyanoacetyl-3,5-dimethylpyrazole and respective amine both they were placed in a reaction flask and dissolved with 40 mL of toluene and heated to reflux for 4 h with stirring. After completing the reaction, the mixture was cooled at room temperature and the solid was filtered and the solvent was recovered. The solid obtain was washed with ethanol and dried. 

*2-cyano-N-phenylacetamide* (**4a**). Gray solid. Yield = 81%, mp 131 °C. ^1^H NMR (400 MHz, DMSO-d_6_) δ 3.89 (s, 2H, CH_2_), 7.09 (t, *J* = 7.4 Hz, 1H, C4 phenyl), 7.33 (t, *J* = 7.9 Hz, 2H, C3 C5 phenyl), 7.54 (d, *J* = 7.9 Hz, 2H, C2 C6 phenyl), 10.28 (s, 1H, NH). ^13^C NMR (101 MHz, DMSO-d_6_) δ 27.18, 116.38, 119.71, 124.37, 129.36, 138.82, 161.45.

*2-cyano-N-(4-methylphenyl)acetamide* (**4b**). Gray solid. Yield = 83%, mp 154 °C. ^1^H NMR (400 MHz, DMSO-d_6_) δ 2.25 (s, 3H, CH_3_ phenyl), 3.86 (s, 2H, CH_2_), 7.13 (d, *J* = 8.4 Hz, 2H, C3 C5 phenyl), 7.42 (d, *J* = 8.4 Hz, 2H, C2 C6 phenyl), 10.19 (s, 1H, NH). ^13^C NMR (101 MHz, DMSO-d_6_) δ 20.90, 27.09, 116.43, 119.72, 129.73, 133.37, 136.33, 161.17.

*2-cyano-N-(4-hydroxyphenyl)acetamide* (**4c**). purple solid. Yield = 84%. mp 229–231 °C. ^1^H MR (400 MHz, DMSO-d_6_) δ 4.44 (s, 2H, CH_2_), 6.35–6.66 (m, 4H, H phenyl), 8.91 (s, 1H, OH). ^13^C NMR (101 MHz, DMSO-d_6_) δ 114.91, 116.90, 119.97, 136.98, 144.44.

#### 3.1.4. General Procedure for the Synthesis of N-Aryl-4,6-dimethyl-2-oxo-1,2-dihydropyridine-3-carbonitrile

Equimolar amounts of acetylacetone and corresponding N-substituted cyanoacetamide (10 mmol) were heated under reflux in a water/ethanol mixture (20 mL) in the presence of a few drops of piperidine as catalyst for 4 h. Product was purified by crystallization from ethanol [29].

*N-phenyl-4,6-dimethyl-2-oxo-1,2-dihydropyridine-3-carbonitrile* (**5a**). White solid. Yield = 85%. mp 255 °C, ^1^H NMR (400 MHz, DMSO-d_6_) δ 2.00 (s, 3H, CH_3_, C6 pyridone), 2.40 (s, 3H, CH_3_ C4 pyridone), 6.36 (s, 1H, H5 pyridone), 7.23–7.32 (m, 2H, C2 C6 phenyl), 7.45–7.60 (m, 3H, C3 C4 C5 phenyl). ^13^C NMR (101 MHz, DMSO-d_6_) δ 20.36, 21.02, 100.38, 108.59, 115.30, 127.79, 128.84, 129.37, 137.46, 151.65, 159.11, 160.28.

*N-(4-methylphenyl)-4,6-dimethyl-2-oxo-1,2-dihydropyridine-3-carbonitrile* (**5b**). White solid. Yield = 78%. mp 274–275 °C. ^1^H NMR (400 MHz, DMSO-d6) δ 2.00 (s, 3H, CH_3_ C6 pyridone), 2.40 (s, 6H CH_3_, 4-methylphenyl, C4 pyridone), 6.34 (s, 1H, H5 pyridone), 7.13 (d, *J* = 7.8 Hz, 2H, C3 C5 phenyl), 7.34 (d, *J* = 7.8 Hz, 2H, C2, C6 phenyl). ^13^C NMR (101 MHz, DMSO-d_6_) δ 20.30, 21.04, 100.30, 108.48, 127.48, 129.85, 134.90, 138.50, 151.88, 158.98, 160.35.

*1-(4-hydroxyphenyl)-4,6-dimethyl-2-oxo-1,2-dihydropyridine-3-carbonitrile* (**5c**) White solid. Yield = 89%, mp 178–179 °C. ^1^H NMR (400 MHz, DMSO-d_6_) δ 1.96 (s, 6H, CH_3_, C4 C6 pyridone), 5.19 (s, 1H, H5 pyridone), 6.75–6.94 (m, 2H, C3 C5 phenyl), 6.98–7.19 (m, 2H, C2 C6 phenyl). ^13^C NMR (101 MHz, DMSO-d_6_) δ 19.96, 29.33, 97.66, 116.23, 116.23, 119.51, 125.47, 126.60, 150.83, 160.48, 194.72.

#### 3.1.5. General Procedure for the Synthesis of N-Alkyl(aryl)-4,6-dimethyl-2-oxo-1,2-dihydropyridine-3-carboxylic acid (**6a–c**)

4.9 mmol of the corresponding N-substituted dihydropyridine-3-carbonitrile were dissolved in 50 mL of distilled water. 49 mmol of KOH were placed in the reaction flask and the mixture was heated to reflux for 24 h with constant stirring. Then the reaction was cooled to room temperature and a HCl solution 1N was added dropwise until a pH equal to 1 was obtained. The mixture formed was placed into separating funnel and extracted with DCM (3 × 20 mL). The combined organic phases were dried and evaporated.

*4,6-dimethyl-2-oxo-1-(4-methylphenyl)-1,2-dihydropyridine-3-carboxylic acid* (**6a**) white solid. Yield = 85% mp 220–222 °C. ^1^H NMR (400 MHz, Chloroform-d) δ 2.09 (s, 3H, CH_3_ C6 pyridone), 2.77 (s, 3H, CH_3_ C4 pyridone), 6.38 (s, 1H, H5 pyridone), 7.14–7.24 (m, 2H, C2 C6 phenyl), 7.47–7.68 (m, 3H, C3 C4 C5 phenyl), 14.94 (s, 1H COOH). ^13^C NMR (101 MHz, Chloroform-d) δ 21.55, 23.63, 113.12, 114.35, 127.12, 129.85, 130.27, 137.12, 149.27, 161.62, 166.07, 166.26.

*4,6-dimethyl-2-oxo-1-(4-methylphenyl)-1,2-dihydropyridine-3-carboxylic acid* (**6b**) white solid. Yield = 95%, 230–232 °C. ^1^H NMR (400 MHz, DMSO-d_6_) δ 2.01 (s, 3H, CH_3_ C6 pyridone), 2.39 (s, 3H, CH_3,_ 4-phenyl), 2.60 (s, 3H, CH_3_ C4 pyridone), 6.65 (s, 1H, H5 pyridone), 7.24 (d, *J* = 8.2 Hz, 2H, C2 C6 phenyl), 7.37 (d, *J* = 8.2 Hz, 2H, C3 C5 phenyl). ^13^C NMR (101 MHz, DMSO-d6) δ 21.19, 21.54, 22.86, 113.14, 113.14, 127.73, 130.65, 135.14, 139.34, 150.99, 159.57, 165.72, 166.15.

*4,6-dimethyl-2-oxo-1-(4-hydroxyphenyl)-1,2-dihydropyridine-3-carboxylic acid* (**6c**) white solid. Yield = 82% 241–243 °C. ^1^H NMR (400 MHz, Chloroform-d) δ 2.34 (s, 3H, CH_3_ C6 pyridone), 2.87 (s, 3H, CH_3_ C4 pyridone), 6.45 (s, 1H, H5 pyridone), 7.24–7.44 (m, 2H, C3 C5 phenyl), 7.57–7.78 (m, 2H, C2 C6 phenyl), 9.77 (s, 1H, OH), 14.98 (s, 1H COOH). ^13^C NMR (101 MHz, Chloroform-d) δ 21.55, 23.63, 112.26, 113.45, 127.12, 129.88, 130.30, 137.10, 149.26, 161.60, 166.08, 169.46.

#### 3.1.6. General Procedure for the Synthesis of the Compound Series **7** and **8**

1.0 mmol of the corresponding N-substituted dihydropyridine-3-carboxylic acid was dissolved in 2 mL of DMF. 2,0 mmol of Diisopropylamine (DIPEA) was added and subsequently 1.2 mmol of BOP reagent was placed in the reaction flask. The mixture was stirred to room temperature for 10 min. Then 1.1 mmol of respective amine was placed in the reaction flask. The mixture of reaction was stirred to room temperature for 2 h until the reaction is complete (monitored by TLC). The mixture was placed, drop by drop, into water-ice bath and solid obtained was filtrated and washed with ethyl ether.

*N-(benzo[d]thiazol-2-yl)-4,6-dimethyl-2-oxo-1-phenyl-1,2-dihydropyridine-3-carboxamide* (**7a**) White solid. Yield = 71%. mp 163–165 °C. ^1^H NMR (400 MHz, Chloroform-d) δ 2.19 (s, 3H: CH_3_, C6 pyridone), 2.89 (s, 3H: CH_3_ C4 pyridone), 6.51 (s, 1H: H5 pyridone), 7.28–8.01 (m, 9H: H aromatics), 12.52 (s, 1H, NH). ^13^C NMR (101 MHz, Chloroform-d) δ 20.08, 21.05, 109.90, 118.62, 119.90, 121.90, 122.82, 125.68, 128.11, 129.15, 130.05, 131.10, 139.31, 145.20, 151.95, 155.27, 160.10, 161.89, 166.58. LS-MS: [M + H] = 376.1.

*N-(1H-benzo[d]imidazol-2-yl)-4,6-dimethyl-2-oxo-1-phenyl-1,2-dihydropyridine-3-carboxamide* (**7b**) White solid. Yield = 77%. mp 177–179 °C. ^1^H NMR (400 MHz, DMSO-d6) δ 1.96 (s, 3H: CH_3_, C6 pyridone), 2.37 (s, 3H: CH_3_, C4 pyridone), 6.27 (s, 1H, H5 pyridone), 7.01–7.35 (m, 9H, H aromatics), 12.54 (s, 2H; NH). ^13^C NMR (101 MHz, DMSO-d6) δ 19.21, 21.69, 109.35, 111.89, 116.31, 120.32, 122.84, 124.76, 129.28, 129.99, 130.40, 137.97, 143.94, 149.29, 150.58, 154.63, 159.74, 162.34, 167.33. LS-MS: [M + H] = 359.2.

*N-(benzo[d]oxazol-2-yl)-4,6-dimethyl-2-oxo-1-phenyl-1,2-dihydropyridine-3-carboxamide* (**7c**) White solid. Yield = 73%. mp 228–230 °C. ^1^H NMR (400 MHz, DMSO-d6) δ 2.34 (s, 3H; CH_3_, C6 pyridone), 2.61 (s, 3H: CH_3_, C4 pyridone), 6.66 (s, 1H, H5 pyridone), 7.47–7.62 (m, 9H, H aromatics), 14.84 (s, 1H, NH). ^13^C NMR (101 MHz, DMSO-d6) δ 20.88, 22.33, 109.81, 112.83, 113.43, 120.98, 127.54, 128.00, 128.54, 129.42, 129.68, 137.23, 138.42, 146.43, 150.20, 151.62, 159.00, 161.40, 167.28. LS-MS: [M + H] = 360.0.

*N-(benzo[d]oxazol-2-yl)-4,6-dimethyl-2-oxo-1-(4-methylphenyl)-1,2-dihydropyridine-3-carboxamide* (**7d**). ^1^H NMR (400 MHz, DMSO-d6) δ 2.37 (s, 3H, CH_3_, 4-phenyl), 2.39 (s, 3H, CH_3_, C6 pyridone), 2.63 (s, 3H, CH_3_, C4 pyridone), 6.57 (s, 1H, H5 pyridone), 7.03–7.42 (m, 8H, H aromatics), 14.98 (s, 1H, NH). ^13^C NMR (101 MHz, DMSO-d6) δ 20.55, 21.42, 22.85, 109.82, 112.85, 113.48, 120.94, 127.71, 128.21, 128.64, 129.40, 135.41, 137.34, 138.44, 146.42, 150.21, 151.64, 159.47, 161.36, 167.57. LS-MS: [M + H] = 373.0.

*N-(1H-benzo[d]imidazol-2-yl)-4,6-dimethyl-2-oxo-1-(4-methylphenyl)-1,2-dihydropyridine-3-carboxamide* (**7e**) White solid. Yield = 79%. mp 198–200 °C. ^1^H NMR (400 MHz, DMSO-d6) δ 1.96 (s, 3H: CH_3_, C6 pyridone), 2.25 (s, 3H, CH_3_ 4-phenyl), 2.37 (s, 3H: CH_3_, C4 pyridone), 6.27 (s, 1H, H5 pyridone), 6.99–7.33 (m, 8H, H aromatics). 12.58 (s, 2H; NH). ^13^C NMR (101 MHz, DMSO-d6) δ 19.21, 21.69, 22.92, 109.35, 111.89, 116.31, 120.32, 122.84, 124.76, 129.28, 129.99, 130.40, 137.97, 143.94, 149.29, 150.58, 154.63, 159.74, 162.34, 167.33. LS-MS: [M + H] = 373.0–374.2.

*N-(1H-benzo[d]imidazol-2-yl)-1-(4-hydroxyphenyl)-4,6-dimethyl-2-oxo-1,2-dihydropyridine-3-carboxamide* (**7f**). White solid. Yield = 72%. mp 176–178 °C. ^1^H NMR (400 MHz, DMSO-d_6_) δ 2.03 (s, 3H, CH_3_, C6 pyridone), 2.16 (s, 3H, CH_3_, C4 pyridone), 6.43 (s, 1H, H5, pyridone), 6.74–7.22 (m, 8H, H aromatics), 9.78 (s, 1H, OH), 12.58 (s, 2H; NH). ^13^C NMR (101 MHz, DMSO-d6) δ 19.34, 22.84, 109.35, 111.89, 116.20, 116.24, 116.31, 120.32, 122.84, 125.01, 125.32, 137.97, 143.94, 149.29, 150.58, 154.63, 156.45, 159.74, 162.34, 167.33. LS-MS: [M + H] = 374.9.

*N-(benzo[d]thiazol-2-yl)-1-(4-hydroxyphenyl)-4,6-dimethyl-2-oxo-1,2-dihydropyridine-3-carboxamide* (**7g**). ^1^H NMR (400 MHz, Chloroform-d) δ 2.02 (s, 3H: CH_3_, C6 pyridone), 2.21 (s, 3H: CH_3_ C4 pyridone), 6.41 (s, 1H: H5 pyridone), 6.71–7.83 (m, 8H: H aromatics), 9.77 (s, 1H, OH), 12.03 (s, 1H; NH). ^13^C NMR (101 MHz, Chloroform-d) δ 20.05, 21.02, 109.85, 118.59, 119.84, 121.89, 122.78, 125.67, 128.09, 129.14, 131.08, 139.29, 145.17, 151.96, 155.30, 160.18, 161.92, 166.64. LS-MS: [M + H] = 391.9–393.9.

*N-cyclohexyl-4,6-dimethyl-2-oxo-1-phenyl-1,2-dihydropyridine-3-carboxamide* (**8a**). White solid. Yield = 78%. mp 139–140 °C. ^1^H NMR (400 MHz, DMSO-d6) δ 1.28–1.76 (m, 10H, CH_2_ cyclohexyl), 1.89 (s, 3H, CH_3_, C6, pyridone), 2.35 (s, 3H, CH_3_, C4 pyridone), 3.83–3.96 (m, 1H, CH cyclohexyl), 6.21 (s, 1H, H5 pyridone), 7.18–7.28 (m, 2H, C2 C6 phenyl), 7.41–7.61 (m, 3H, C3 C4 C5 phenyl), 8.84 (d, *J* = 7.8 Hz, 1H, NH). ^13^C NMR (101 MHz, DMSO-d6) δ 20.74, 20.86, 24.23, 25.64, 33.10, 49.38, 109.97, 120.77, 127.98, 128.54, 129.43, 138.38, 146.25, 151.64, 161.58, 164.01. LS-MS: [M + H] = 324.6.

*N-cycloheptyl-4,6-dimethyl-2-oxo-1-phenyl-1,2-dihydropyridine-3-carboxamide* (**8b**). White solid. Yield = 80%. mp 138–140 °C. ^1^H NMR (400 MHz, DMSO-d6) δ 1.34–1.77 (m, 12H, CH_2_ cycloheptyl), 1.89 (s, 3H, CH_3_, C6, pyridone), 2.34 (s, 3H, CH_3_, C4 pyridone), 3.82–3.96 (m, 1H, CH cycloheptyl), 6.25 (s, 1H, H5 pyridone), 7.19–7.28 (m, 2H, C2 C6 phenyl), 7.41–7.61 (m, 3H, C3 C4 C5 phenyl), 8.85 (d, *J* = 7.8 Hz, 1H, NH). ^13^C NMR (101 MHz, DMSO-d6) δ 20.74, 20.86, 23.67, 27.66, 34.20, 49.31, 109.97, 120.77, 127.98, 128.54, 129.43, 138.38, 146.25, 151.64, 161.58, 164.01. LS-MS: [M + H] = 339.1–341.1.

*N-(adamantan-1-yl)-4,6-dimethyl-2-oxo-1-phenyl-1,2-dihydropyridine-3-carboxamide* (**8c**). White solid. Yield = 83%, mp 164–165 °C. ^1^H NMR (400 MHz, DMSO-d6) δ 1.65–1.87 (m, 14H, H adamantyl), 1.94 (s, 3H, CH_3_, C6, pyridone), 2.38 (s, 3H, CH_3_, C4 pyridone), 6.24 (s, 1H, H5 pyridone), 7.08–7.43 (m, 4H, H phenyl), 7.91 (s, 1H, NH). ^13^C NMR (101 MHz, DMSO-d6) δ 20.74, 20.86, 29.75, 36.10, 42.38, 51.32, 109.97, 120.77, 127.97, 128.53, 129.43, 138.38, 146.25, 151.63, 161.58, 164.01. LS-MS: [M + H] = 377.2.

*N-(adamantan-1-yl)-4,6-dimethyl-2-oxo-1-(4-methylphenyl)-1,2-dihydropyridine-3-carboxamide* (**8d**). White solid. Yield = 71%. mp 154–155 °C. ^1^H NMR (400 MHz, DMSO-d6) δ 1.47–1.73 (m, 14H, H adamantyl), 1.96 (s, 3H, CH_3_ C6 pyridone), 2.38 (s, 6H CH_3_ 4-phenyl, C4 pyridone), 6.41 (s, 1H, H5 pyridone), 7.09 (d, *J* = 8.2 Hz, 2H, C2 C6 phenyl), 7.32 (d, *J* = 8.2 Hz, 2H, C3 C5 phenyl), 8.56 (s, 1H, NH). ^13^C NMR (101 MHz, DMSO-d6) δ 20.65, 20.86, 28.80, 36.05, 41.06, 50.71, 109.98, 121.30, 127.67, 129.86, 135.82, 137.93, 146.20, 151.33, 161.68, 164.40. LS-MS: [M + H] = 391.1.

*N-cycloheptyl-4,6-dimethyl-2-oxo-(4-methylphenyl)-1,2-dihydropyridine-3-carboxamide* (**8e**). White solid. Yield = 74%. mp 238–240 °C. ^1^H NMR (400 MHz, DMSO-d6) δ 1.34–1.77 (m, 12H, CH_2_ cycloheptyl), 1.94 (s, 3H, CH_3_, C6, pyridone), 2.33 (s, 3H CH_3_ 4-phenyl)2.37 (s, 3H, CH_3_, C4 pyridone), 3.45–3.65 (m, 1H, CH cycloheptyl), 6.26 (s, 1H, H5 pyridone), 7.20–7.29 (m, 2H, C2 C6 phenyl), 7.40–7.65 (m, 3H, C3 C4 C5 phenyl), 7.92 (s, 1H, NH). ^13^C NMR (101 MHz, DMSO-d6) δ 20.74, 20.86, 29.75, 36.10, 42.38, 51.32, 109.97, 120.77, 127.97, 128.53, 129.43, 138.38, 146.25, 151.63, 161.58, 164.01. LS-MS: [M + H] = 353.5.

*N-cycloheptyl-1-(4-hydroxyphenyl)-4,6-dimethyl-2-oxo-1,2-dihydropyridine-3-carboxamide* (**8f**). White solid. Yield = 81%. mp 232–233 °C. ^1^H NMR (400 MHz, DMSO-d6) δ 1.22–1.77 (m, 12H, H cycloheptyl), 1.90 (s, 3H, CH_3_ C6 pyridone), 2.33 (s, 3H, CH_3_ C4 pyridone), 6.21 (s, 1H, H5 pyridone), 6.86 (d, *J* = 8.7 Hz, 2H, C2 C6 pyridone), 6.99 (d, *J* = 8.7 Hz, 2H, C3 C5 pyridone), 8.91 (d, *J* = 7.8 Hz, 1H, NH), 9.75 (s, 1H, OH). ^13^C NMR (101 MHz, DMSO-d6) δ 20.82, 20.93, 23.66, 27.66, 34.20, 49.25, 109.87, 115.81, 120.41, 128.81, 129.42, 147.05, 151.58, 157.30, 161.92, 164.12. LS-MS: [M + H] = 354.9.

*N-cyclohexyl-1-(4-hydroxyphenyl)-4,6-dimethyl-2-oxo-1,2-dihydropyridine-3-carboxamide* (**8g**) White solid. Yield = 78%. mp 139–140 °C. ^1^H NMR (200 MHz, DMSO-d_6_) δ 1.01–1.79 (m, 10H, CH cyclohexyl), 1.90 (s, 3H, CH_3_ C6 pyridone), 2.33 (s, 3H, CH_3_ C4 pyridone), 3.57–3.85 (m, 1H, CH cyclohexyl), 6.21 (s, 1H, H5 pyridone), 6.86 (d, *J* = 8.7 Hz, 2H, C2 C6 phenyl), 6.99 (d, *J* = 8.7 Hz, 2H, C3 C5 phenyl), 8.87 (d, *J* = 7.8 Hz, 1H, NH), 9.76 (s, 1H, OH). ^13^C NMR (101 MHz, DMSO-d6) δ 20.82, 20.93, 24.31, 25.21, 32.24, 47.05, 109.87, 115.81, 120.39, 128.81, 129.41, 147.08, 151.61, 157.30, 161.89, 164.36. LS-MS: [M + H] = 340.6.

### 3.2. Human CB2R (h) (Agonist Effect) Cellular Functional Assay

Functional evaluation was carried out by Eurofins Cerep service assay: CB2 Human Cannabinoid GPCR Cell Based Agonist cAMP Assay [24]. CHO cells stably expressing hCB2R (Eurofins Cerep) are suspended in HBSS buffer (Invitrogen) complemented with 20 mM HEPES (pH 7.4), then distributed in microplates at a density of 7.5 × 10^3^ cells/well in the presence of either of the following: HBSS (basal control), the reference agonist at 100 nM (stimulated control) or various concentrations (EC50 determination), or the test compounds. Thereafter, the adenylyl cyclase activator NKH 477 is added at a final concentration of 3 μM. Following 10 min incubation at 37 °C, the cells are lysed, and the fluorescence acceptor (D2-labeled cAMP) and fluorescence donor (anti-cAMP antibody labeled with europium cryptate) are added. After 60 min at room temperature, the fluorescence transfer is measured at λ_ex_ = 337 nm and λ_em_ = 620 and 665 nm using a microplate reader (Envison, Perkin Elmer). The cAMP concentration is determined by dividing the signal measured at 665 nm by that measured at 620 nm (ratio). The results are expressed as a percent of the control response to 100 nM WIN55212-2. The standard reference agonist is WIN 55212-2, which is tested in each experiment at several concentrations to generate a concentration-response curve from which its EC50 value is calculated.

### 3.3. Cell Culture

Cell lines were grown in culture media supplemented with 10% fetal bovine serum (FBS), 100 U/mL of penicillin, 100 μg/mL of streptomycin and 1% of non-essential aminoacids. Cells between passage 5 to 10 were used in experiments. Dulbecco’s Modified Eagle Medium (DMEM) was used for HepG2 cells and Roswell Park Memorial Institute (RPMI) was used for HL-60 cells. All cells were maintained in a humidified incubator at 37 °C and 5% CO_2_ saturation. All cellular lines used were provided by Dr. Mario Faúndez, Laboratory of Molecular Pharmacology, Faculty of Chemistry and Pharmacy, Pontificia Universidad Católica de Chile.

### 3.4. Cell Viability

The effect of compounds in cell viability was determined by Neutral Red uptake assay. Cells were seeded in 100 μL of media at a density of 10^4^ cells/well in 96-well microtiter plates. Solutions of compounds were previously prepared in DMSO and 1 μL of the corresponding solution was added to each well. The final volume of each well was adjusted to 200 μL. After 72 h of incubation, culture media was removed and 100 μL of 10 μg/mL neutral red solution prepared in culture media was added to each well and incubated for 3 h. Then, media was aspirated, the plate was washed three times with PBS 1X and 100 μL of neutral red distain solution (50:49:1 ethanol:water:glacial acetic acid) was added. The plate was placed for 15 min in a shaker and fluorescence was measured using Cytation 5 apparatus (Biotek, Winooski, VT, USA) at 530/645 nm excitation/emission wavelengths.

### 3.5. Molecular Docking

The cryoEM structure of the CB2 receptor co-crystallized with agonist WIN-55212-2 (PDB:6PT0) [23] was used for this work, obtained from the RCSB Protein Data Bank. The structure was prepared for docking using Discovery Studio Visualizer 2020 [30], removing all molecules except the receptor itself. The analyzed compounds were submitted to energy minimization using MMFF94 forcefield [31] using OpenBabel [32], embedded in the PyRx software [33]. Docking calculations were carried in the same software using AutoDock Vina [34] with a 25 × 25 × 25 Å box centered between Val113 and Phe183 and with an exhaustiveness of 8. Docking results were visualized and analyzed with Discovery Studio Visualizer 2020. Protein superimposition between PDB:6PT0 and the crystal structure of CB2 receptor co-crystallized with selective agonist AM12033 (PDB:6KPC) [26] was done with UCSF Chimera software [35].

## 4. Conclusions

In this study we report a new and non-toxic CB2R agonist structurally based on the 2-pyridone moiety. Our findings provide further evidence to support a role of the 2-pyridone ring as bridging scaffold suitable to effectively orient substituent groups towards the main described pockets of the CB2R orthosteric site. The derived SAR suggests that bulky substitutions in the carboxamide moiety are advantageous for agonist activity with the spatial arrangement being key to allow an effective accommodation into the orthosteric pocket sub cavities.

The adamantyl group bound to the pyridone ring in **8d** highlights the importance of a hydrophobic and voluminous moiety that can be accommodated inside cavity 1. Additionally, hydrophobic substituents bound to the nitrogen atom of the pyridone ring such as a tolyl group is appropriate to interact with residues in the cavity 2. The predicted binding mode is consistent with the agonist profile observed in functional assays and provides information on the role of the pyridone substructure in cannabinoid ligands. Furthermore, the synthetized compounds were non-toxic in different tumor cell lines with compound **8d** inhibiting the proliferation in HepG2 and HL-60 cells.

Even though previous reports identify pyridone-based derivatives as selective CB2R/CB1R ligands [17,18,19] future studies to assess the selectivity profile and ADME properties of compound **8d** are necessary. Additionally, further studies are also needed to identify the mechanism linked to the anti-proliferative effect of this compound and its potential pharmacological application.

Together these results encourage to gain better understanding of the cannabinoid receptor binding site and further explore the 2-pyridone scaffold in the development of new CB2R ligands with potential therapeutical application for the treatment of pain and inflammation related disorders in both the peripheral and CNS [11,12,36,37].

## Data Availability

Not applicable.

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
