# Peer review of "New Pyridone-Based Derivatives as Cannabinoid Receptor Type 2 Agonists"

_ijms, 2021, doi:10.3390/ijms222011212_

Round 1

Reviewer 1 Report

  1. Introduction can be extended with more details regarding the target of cannabinoids and their potential therapeutical effects, see: doi: 10.3390/jpm11060494, doi: 10.3390/jcm9082395
  2. Add the strengths and limitations of this study and how these findings can be applied in practice.

Author Response

Point 1: Introduction can be extended with more details regarding the target of cannabinoids and their potential therapeutical effects, see: doi: 10.3390/jpm11060494, doi: 10.3390/jcm9082395

Response 1: We thank Reviewer 1 for his corrections. We have extended the Introduction section with more details according to the Reviewers’ comments and suggested references (lines 41-58).

Point 2: Add the strengths and limitations of this study and how these findings can be applied in practice.

Response 2: We thank Reviewer 1 for his corrections. We have included strengths, limitations, and further applications in the Conclusion section according to the Reviewers’ suggestions (lines 498-506).

Reviewer 2 Report

The manuscript provides new data on synthesis and biological activity of 2-pyridone derivatives, particularly 8d compound which exibits similar agonist potency to the endogenous anandamine and 2-AG. Since the synthesized compounds exert agonist activity of human cannabinoid receptor type II, their translational value to treat neuropatic pain is high. I recommend the manuscript to be published in the Journal. However, some corrections are necessary (e.g., in chapter Materials and Methods: lines 409, 433; instead of 7.5x103 should be 7.5x10(3 in superscript) etc.,

Author Response

Point 1: The manuscript provides new data on synthesis and biological activity of 2-pyridone derivatives, particularly 8d compound which exibits similar agonist potency to the endogenous anandamine and 2-AG. Since the synthesized compounds exert agonist activity of human cannabinoid receptor type II, their translational value to treat neuropatic pain is high. I recommend the manuscript to be published in the Journal. However, some corrections are necessary (e.g., in chapter Materials and Methods: lines 409, 433; instead of 7.5x103 should be 7.5x10(3 in superscript) etc.,

Response 1: We thank Reviewer 2 for his corrections. We have corrected superscripts in lines 409 (435 in the corrected manuscript) and 433 (460 in corrected the manuscript) as indicated by the Reviewer 2 together with additional typographical or style errors in lines 63, 81, Table 1, 183,196, 197, 211 etc.

Reviewer 3 Report

The work titled: New Pyridone-Based Derivatives as Cannabinoid Receptor Type 2 Agonists is a well-written work. It contains all the elements of a research work. In my opinion, the summary of the work is too poor, and this is what gives it the greatest value. It lacks the specifics that it brings to the world of science and any plans for further research.

Author Response

Point 1: The work titled: New Pyridone-Based Derivatives as Cannabinoid Receptor Type 2 Agonists is a well-written work. It contains all the elements of a research work. In my opinion, the summary of the work is too poor, and this is what gives it the greatest value. It lacks the specifics that it brings to the world of science and any plans for further research.

Response 1: We thank Reviewer 3 for his corrections. We have modified the abstract section according to the reviewers’ suggestions (lines 12 to 25) including relevance and therapeutical application.